# Amylase and Cellulase Production from Newly Isolated *Bacillus subtilis* Using Acid Treated Potato Peel Waste

**DOI:** 10.3390/microorganisms12061106

**Published:** 2024-05-29

**Authors:** Qudsia Mushtaq, Uzair Ishtiaq, Nicolas Joly, Javed Iqbal Qazi, Patrick Martin

**Affiliations:** 1Microbial Biotechnology Laboratory, Department of Zoology, University of Punjab, New Campus, Lahore 54590, Pakistan; qazi.zool@pu.edu.pk; 2Department of Research and Development, Paktex Industries, 2.5 KM Tatlay Road, Kamoke 52470, Pakistan; uzair.saroya9@gmail.com; 3Department of Life Sciences, University of Management and Technology, Lahore 54770, Pakistan; 4ULR7519—Unité Transformations and Agro-Resources, University Artois, UniLasalle, F-62408 Béthune, France; nicolas.joly@univ-artois.fr

**Keywords:** cellulase, amylase, Box–Behnken design, proximate analysis, potato peel waste (PPW), acidic treatment, *Bacillus subtilis*

## Abstract

Species belonging to the genus *Bacillus* produce many advantageous extracellular enzymes that have tremendous applications on a commercial scale for the textile, detergent, feed, food, and beverage industries. This study aimed to isolate potent thermo-tolerant amylolytic and cellulolytic bacterium from the local environment. Using the Box–Behnken design of response surface methodology, we further optimized the amylase and cellulase activity. The isolate was identified by 16S rRNA gene sequencing as *Bacillus subtilis* QY4. This study utilized potato peel waste (PPW) as the biomaterial, which is excessively being dumped in an open environment. Nutritional status of the dried PPW was determined by proximate analysis. All experimental runs were carried out in 250 mL Erlenmeyer flasks containing acid treated PPW as a substrate by the thermos-tolerant *Bacillus subtilis* QY4 incubated at 37 °C for 72 h of submerged fermentation. Results revealed that the dilute H_2_SO_4_ assisted autoclaved treatment favored more amylase production (0.601 IU/mL/min) compared to the acid treatment whereas high cellulase production (1.269 IU/mL/min) was observed in the dilute acid treatment and was found to be very effective compared to the acid assisted autoclaved treatment. The *p*-value, F-value, and coefficient of determination proved the significance of the model. These results suggest that PPW could be sustainably used to produce enzymes, which offer tremendous applications in various industrial arrays, particularly in biofuel production.

## 1. Introduction

The increasing amount of food waste worldwide is becoming an alarming problem for waste management plants and has amounted to 1.3 million tons/year. Huge amounts of these food wastes possess severe environmental concerns due to decomposition. The food processing industry is one of the most important businesses that produces a huge amount of organic waste that needs to be managed appropriately to avoid environmental pollution but can also contribute toward an economic boost from the utilization of by-products [1]. More than 1.3 billion tons of food is wasted, which is equal to almost 13.8% of the total food production, with significant losses at the final consumption stages and in households [2].

The potato is considered as the most important food crop after wheat, rice, and maize [3]. The worldwide production of potatoes reached up to 361.09 million metric tons (MMT) in 2013 and increased to 370.43 MMT in 2019 [4]. China is the greatest producer of potatoes followed by India, Russia, Ukraine, and the United States of America [5]. According to the United Nations Food and Agricultural Organization (FAO), it is estimated that around 8000 kilotons of PPW might be produced in 2030, with related greenhouse gas emissions of 5 million tons [6].

PPW is the main waste product of potato-processing industries and is generated in huge quantities. However, due to its composition, availability, and zero cost, it might be used as a renewable resource for the production of value-added products [7]. Moreover, other plants including the potato starch, flour, and canning industries are also responsible for huge amounts of PPW, whose disposal increases environmental concerns [8,9]. These industries generate 70 to 140 thousand tons of PPW on annual basis [10]. Most of this waste is discarded through landfills, with the accompanying environmental pollution [11,12]. To overcome this global issue, the appropriate management of this waste into value added products can therefore be advantageous, not only to the food industry, but also to policymakers, decreasing the environmental impacts of these industries [10].

The biotechnological and eco-friendly applications of enzymes from microorganisms have drawn a great deal of attention from various researchers worldwide. Amylases and cellulases are highly potent enzymes for fermentation and also offer tremendous applications in various industrial arrays due to their prospective thermal and pH stability [13,14]. Amylases characterize approximately 30–33% of the world enzyme market. They have widespread occurrence and have the maximum market share of enzyme sales. These encompass hydrolases, which break down starch into various products such as dextrins and eventually glucose units [15,16]. The key areas of industry where amylases and cellulases are increasingly being applied are in textile, healthcare, pulp, paper, detergent, food, beverages, pharmaceutical, biofuel, and pretreatment methods [17,18,19,20,21,22]. Various research reports have investigated the potential of PPW enzyme production and various other value-added products [7,23,24,25,26].

The present study aimed to provide a sustainable and cost-effective production of bacterial amylase and cellulase by employing starch rich biowaste PPW. PPW containing a high content of various nutritional components such as carbohydrates of up to 65% has been effectively used for the production of value-added products. Pretreated PPW was utilized as the carbon source by locally isolated bacterium Bacillus subtilis QY4, which liberates a high content of amylase and cellulase. These enzymes are in high demand in various industrial sectors and can be excessively used for substrate pretreatment processes prior to fermentation. Our study reports the high yield of low-cost amylase and cellulase, which can be used for different industrial and biotechnological applications.

## 2. Materials and Methods

### 2.1. PPW Characterization

The compositional analysis of PPW was characterized by determining the content of different components including moisture, ash, crude fat, crude protein, crude fiber, and carbohydrates using the standard protocols of AOAC [27].

### 2.2. Isolation of Bacterial Strains

Different bacterial strains with amylase and cellulase producing potential were isolated from the local environment. Samples were collected from waste, discarded, and spoiled fruits taken from different vendor shops located close to the University of Punjab, Lahore, Pakistan. Samples were taken in sterilized vials and transported to the Microbial Biotechnology Laboratory at the University of Punjab for processing. Measurements of pH and temperature of the original locations were carried out.

### 2.3. Microbiological Quality Control

The microbiological quality control was maintained by ensuring the growth of pure culture. Pure culture was obtained by cross streaking on nutrient agar plates, followed by repeated streaking. This procedure was performed until the pure culture was maintained. To minimize human errors, streaking was performed in triplicate. The inoculating loop used for inoculation was heated at an elevated temperature using an open flame to ensure the killing of any microbial growth. Ultraviolet lamps were turned on in a laminar flow hood for 15 min to ensure the removal of any microbial growth. The whole procedure was performed in an ISO certified cleanroom.

### 2.4. Culture Media and Isolation of Thermo-Tolerant Bacterium

The samples were serially diluted and then inoculated on nutrient agar (Peptone 5 g, yeast extract 1.5 g, beef extract 1.5 g sodium chloride 5 g, agar-agar 15 g/L, pH 7.4 ± 0.2 at 25 °C). Different colonies were selected from general-purpose medium nutrient agar. Isolated colonies were further grown on the selective medium. All of the cultures were raised at 37 °C for 24 h to 48 h. The selective medium for the screening of amylolytic and cellulolytic bacteria comprised K_2_HPO_4_, KH_2_PO_4_, NaCl, MgSO_4_.7H_2_O, (NH_4_)_2_SO_4_, yeast extract, agar, and starch (for amylase)/carboxy methyl cellulose (CMC for cellulase). The self-constructed media comprised 5% PPW, and 1% yeast extract was also used for the production of crude enzymes (i.e., amylase and cellulase). To check the pH tolerance of the isolated bacterium, the sterile nutrient broth was prepared with different pH such as 6, 6.5, 7.0, 7.5, and 8.0, and after inoculation, were incubated at 37 for 72–96 h. To check the thermo-stability of the isolated bacterium, the culture temperature was maintained at various values such as 37, 40, 50, and 60 °C, with shaking maintained at 150 rpm for 72–96 h. The growth of bacterium was checked at 600 nm using a light spectrophotometer. To preserve the bacterium, glycerol stocks were prepared with bacterial culture and preserved at −40 °C.

### 2.5. Amylolytic Potential

The amylase potential of bacterial strains was checked on both the self-designed and referenced selective media. The selective medium, as described in Table 1, for amylase production was prepared, autoclaved, poured in Petri plates of 100 mm diameter and allowed to solidify. After 24 h, these plates were inoculated with growth of the selected strain QY4 (PP439596). These inoculated plates were incubated at 37 °C for 24 h. Afterward, the starch hydrolysis test was performed with Lugol’s iodine solution following the method of Lal and Cheeptham [28], where 3 mL of iodine solution was poured on the 100 mm diameter of the Petri plate to visualize the transparent zone indicating the extracellular amylase production by the isolated bacterium QY4 (PP439596).

### 2.6. Cellulolytic Potential

The production of extracellular cellulase on the CMC agar plates was checked following the method of Gohel [29]. The selective medium, as described in Table 1 for cellulase production, was prepared, autoclaved, poured in Petri plates, and allowed to solidify. After 24 h, these plates were inoculated with growth of the selected strain. These inoculated 100-mm diameter Petri plates were incubated at 37 °C for 24 h. Afterward, these plates were flooded with 2 mL of 0.1% (*w*/*v*) Congo red solution for 20 min, and washed with 3 mL of 1 M NaCl for 25 min. The cellulose degrading zones were observed around bacterial growth, confirming that the strain could hydrolyze cellulose.

### 2.7. Substrate Preparation

Discarded and rejected PPWs were obtained from a local fries shop (Gate No. 4, Quaid-e-Azam Campus, University of Punjab, Lahore, Pakistan). These were properly washed to remove all dirt, sun dried for 24 h, and then oven dried at 60 °C for 72 h until constant weight was attained. These were then powdered (50 mm) by an electric mill. The pulverized material was stored in airtight containers at room temperature (25 °C ± 5 °C) until further use.

### 2.8. DNA Extraction and Storage

To extract genomic DNA, 3 mL of the incubated culture was centrifuged and the resulting pellet was suspended in lysis solution and the supernatant stored in Eppendorf tubes, which were subsequently stored at −40 °C until further use. To assess the integrity of the DNA, electrophoresis was performed using 0.5% agarose gel containing EtBr in 0.5× TAE buffer.

### 2.9. Taxonomic Identification of Bacterial Isolate

The bacterium was isolated by repeated streaking on the nutrient agar medium and stored at 4 °C. 16S rRNA gene sequencing identified it, and a detailed protocol of molecular identification has been described in an earlier report [30].

The 16S rRNA gene was amplified from the extracted DNA by polymerase chain reaction using 27F 5′ (AGA GTT TGA TCM TGG CTC AG) 3′ and 1492R 5′ (TAC GGY TAC CTT GTT ACG ACT T) 3′. The amplicons were cloned and the insert was sequenced from Macrogen (Seoul, Republic of Korea). The sequence obtained was submitted in GenBank on 6 March 2024 and aligned using CLUSTAL W 1.81. The phylogenetic tree of Bacillus subtilis QY4 (NR_104873.1) was constructed using the neighbor-joining method in MEGA 5.0 (Molecular Evolutionary Genetics Analysis, version 5.0) software [31].

### 2.10. Sequencing Primer Information

For the sequencing of QY4 (PP439596), the sequencing primer name was 785F and its sequence was 5′ (GGA TTA GAT ACC CTG GTA) 3′ and 907R 5′ (CCG TCA ATT CMT TTR AGT TT) 3′, respectively. Meanwhile, the PCR name and sequence was 27F 5′ (AGA GTT TGA TCM TGG CTC AG) 3′ and 1492R 5′ (TAC GGY TAC CTT GTT ACG ACT T) 3′, respectively.

### 2.11. Acidic Pretreatment of PPW

Pretreatment was carried out in 250 mL Erlenmeyer flasks with a working volume of 100 mL. The PPW concentrations were varied according to the experimental design (Table 2) and soaked in different concentrations of H_2_SO_4_ for various time periods. The experiment was conducted using two methods (i.e., acidic and acid assisted autoclaved treatment). Following the soaking process, the latter category of the pretreatment reactors was subjected to pressurized heat at 121 °C (21 psi) for 15 min. Following the treatment process, all samples were filtered and the remaining solid sample was taken, neutralized with 1 M NaOH, and kept in an incubator at 70 °C until a consistent weight was achieved. This residual substrate was further investigated to determine the amylase and cellulase content.

### 2.12. Production Media Used for Fermentation

Acid treated and acid assisted autoclaved treated PPW were used as the substrate to obtain optimum amylase production. The sulfuric acid-treated PPW was neutralized with 1 M NaOH and homogenized with a mortar and pestle to obtain a uniform size. For amylase production, 250 mL flasks were used with 100 mL of the PPW medium comprising 2% pretreated PPP and 1% yeast extract with pH 5.0. The flasks containing media were autoclaved and allowed to cool down. The media were sterilized and inoculated with a 2% inoculum of a 24 h-old culture of *Bacillus subtilis* QY4 and incubated at 37 °C at 120 rpm for 72 h. Samples were taken after fermentation for 72 h and then centrifuged at 10,000 rpm for 10 min at 4 °C. The clear supernatant without bacterial growth was examined as the crude enzyme source and was used for the amylase and cellulase assay. The same protocol was followed for all 17 runs of the Box–Behnken design (BBD), and the experiments were performed in triplicate.

### 2.13. Amylase and Cellulase Assays

The amount of indigenously produced amylase was determined according to the method of Bernfeld [32]. Different series of maltose solutions (0–100 mg/L) were prepared to establish a standard curve. Then, 0.5 mL of a diluted sample was taken in a test tube with 0.5 mL of 1% starch soluble in 0.05 M phosphate buffer (pH 5.5) and incubated at 40 °C for 30 min. The reaction mixture was arrested using 3 mL of DNS. One enzyme unit was defined as the amount of enzyme that released 1 µmol of glucose per mL per min.

The cellulase content was determined based on the method reported [33]. Different series of glucose solutions (0–100 mg/L) were prepared to establish a standard curve. Then, 0.5 mL of the diluted sample was taken in a test tube with 0.5 mL of 1% CMC soluble in 0.05 M sodium citrate buffer pH 5.5 and incubated at 50 °C for 30 min. The reaction mixture was arrested using 3,5-dinitrosalicylic acid. The cellulase unit was accordingly defined as the amount of enzyme that released 1 µmol of glucose per mL per min. All of the experiments were carried out in triplicate.

### 2.14. Experimental Design

In the present investigation, three independent variables (i.e., substrate conc., H_2_SO_4_ conc., and time were symbolized as X_1_, X_2_ and X_3_) were optimized for amylase production. Different time periods were selected for the acid treatment of PPW, and afterward, the acid hydrolyzed PPW was used by the microbial strain to produce crude enzymes. BBD was used for optimization; coded values of all parameters are mentioned in Table 2. The mathematical relationship among variables was calculated by the second-order polynomial equation. The response (i.e., enzyme production) was calculated from the following equation using Minitab software 16.
Y = β_0_ + Σ β_i_X_i_ + Σβ_ii_X_i_^2^ + Σβ_ij_X_j_X_j_(1)
where Y is the response, β_0_ is the intercept, β_i_ is the linear coefficient, β_ii_ is a squared coefficients, and β_ij_ is the interactive coefficients.

### 2.15. Deposition of Strain

The strain *Bacillus subtilis* QY4 (PP439596) isolated in this study was deposited in Microbial Stock at the Microbial Biotechnology Laboratory located at the Institute of Zoology, University of Punjab, Lahore, Pakistan and Microbial Stock Conservation at the Department of Research and Development at Paktex Industries, with conservation numbers MBLZPU-9010 and R&D/PI202-1, respectively.

## 3. Results and Discussion

### 3.1. Proximate Composition of Dried PPW

Proximate analysis of the raw PPW revealed that it contained 10.29% moisture, 8.81% ash, 0.44% crude fat, 15.45% crude protein, 4.4% crude fiber, and 65.01% carbohydrates on dry weight basis, as shown in Table 3. A protein content of 10.54% in dried potato peels has been reported [34,35]. Another research report has recorded an almost similar level of fat from the potato peels, which was 0.28% [36]. The moisture, fiber, and ash values almost agreed with those reported by Rowayshed [37], while the carbohydrate content value of PPW was similar to the value reported by Badr SA and El-Wasif [38]. The notably higher amounts of carbohydrate and protein rendered the waste biomass a rich medium encompassing the main nutrients essential for microorganism growth and amylase production. In addition, this waste also contained a rich amount of starch, which could serve as an efficient feedstock for biofuel production.

### 3.2. Cultivation and Isolation of Bacterium

Various species of Bacillus secrete profitable extracellular enzymes that have remarkable applications in various industrial sectors including food, feed, textile, detergent, and beverages. The application of molecular and biological methods to bring an improvement in strains is being vigorously investigated. *Bacillus* species are lucrative industrial organisms due to their short lifespan, extracellular proteins, and welfare for humans. In this work, we studied the most important enzymes i.e., amylase and cellulase produced by *Bacillus* species [39]. In this study, a thermo-tolerant bacterial strain was isolated from the fruit waste. The temperature and pH ranges of the sampling sites were recorded between 30 and 45 °C and pH 6.5–8.5, respectively. The bacterium was isolated on nutrient agar and its potential was also checked on indigenous medium. In total, twenty bacterial strains were isolated, but further screening was conducted on the basis of morphological differences, and the amylase and cellulase producing potential was confirmed on selective media comprising starch and CMC. *Bacillus subtilis* QY4 PP439596 was grown on nutrient agar plates, incubated at 37 °C for 48–72 h, maintained at 4 °C, and sub-cultured at 4-week intervals. The growth patterns of the colony showed diversity in response to environmental conditions such as nutrient and agar concentration. The colony was observed as creamy white, medium size, with a circular outline. The colony size was variable according to incubation time; a colony size of 10–14 mm of *Bacillus subtilis* QY4 (PP439596) was observed after 72–96 h of incubation at 37 °C on nutrient agar media. The cultivation results indicate that the selected strain exhibited proliferation within a temperature range of 40–60 °C. This characteristic classifies the isolates as thermo-tolerant. The optimal pH range for proliferation was generally found between 6 and 8, with a dominant pH optimum between 6 and 7. Observations under an optical microscope showed that the isolated strain corresponded to Gram-positive bacilli, as shown in Table 4.

### 3.3. Amylolytic and Cellulolytic on Indigenous Media PPW

Acid treated PPW was used as a substrate by the bacterial strain to enhance the production of amylase and cellulase. The amylolytic and cellulolytic potential of *B. subtilis* was checked on selective medium, as reported in Table 1. The sizes of the bacterial colony, hydrolysis zone (HZ), and hydrolysis zone index (HZI) of amylase production were measured as 14 mm, 9.0 mm, and 1.64, respectively, whereas for cellulase production, these sizes were measured as 10 mm, 8.0 mm, and 1.8, respectively, using *Bacillus subtilis*. The ability of the bacterium to grow and produce amylase by utilizing commercial media containing starch is depicted in Figure 1 and Figure 2.

### 3.4. Molecular Identification of the Bacterial Isolate; B. subtilis *QY4*

The bacterial strain isolated from the local environment was identified as *Bacillus subtilis*. Its length was calculated as 1538 base pairs. The phylogenetic tree of *Bacillus subtilis* QY4 (NR_104873.1), as depicted in Figure 3, was constructed using the neighbor-joining method, which described the similarity index of the bacterial isolate to other species. The isolated bacterium showed 99% homology to *Bacillus subtilis* strain ST15 (MK511833.1) and *Bacillus subtilis* strain ZB (KX450400.1).

### 3.5. Effect of Pretreatment Conditions on Amylase and Cellulase Productions

In this study, different pretreatment conditions were used to maximize amylase and cellulase production in submerged fermentation by *B. subtilis* using PPW as a cheap biowaste. BBD was conducted using three independent variables such as substrate H_2_SO_4_ conc. and time.

The observed (Obs.) and predicted (Pred.) values of amylase and cellulase content after dilute H_2_SO_4_ and H_2_SO_4_ assisted autoclaved treatment are shown in Table 5. Multiple regression analysis was applied and second-order polynomial regression equations Equations (2)–(5) revealed the influence of pretreatment conditions on amylase production using PPW.
Amylase (IU/mL) after acidic treatment = 0.840 − 0.1724X1 + 1.1098X2 − 0.1916X3 + 0.006742X_1_^2^ − 0.651X_2_^2^+ 0.01114X_3_^3^ + 0.0179X1*X2 + 0.00367X1*X3 + 0.0169X2*X3(2)
Amylase (IU/mL) after acid assisted autoclaving treatment = −5.12 + 0.3139X1 + 5.80X2 + 0.285X3 − 0.00833 X_1_^2^ − 2.748X_2_^2^ − 0.04163X_3_^3^ − 0.1368X1*X2 − 0.00330X1*X3 + 0.2798X2*X3(3)
Cellulase (IU/mL) after acidic treatment = 4.5779 − 0.46227X1+ 3.20563X2 − 1.08007X3 + 0.01465X_1_^2^ − 0.79563X_2_^2^ + 0.10836X_3_^3^ + 0.06750X1*X2 + 0.01619X1*X3- 0.42463X2*X3(4)
Cellulase (IU/mL) acid assisted autoclaving treatment = 1.32150 + 0.0562X1 − 1.50833X2 − 0.18675X3 − 0.00309X_1_^2^ + 0.40729X_2_^2^ − 0.00305X_3_^3^ − 0.01075X1*X2 + 0.00357X1*X3+ 0.21188X2*X3(5)
where X1, X2, X3 represent linear coefficients, X_1_^2^, X_2_^2^, X_3_^2^ represent squared coefficients, and X1*X2, X1*X3, X2*X3 represent interactive coefficients.

Acid assisted autoclaved pretreatment proved to be more effective for amylase production and yielded up to 0.6010 IU/mL/min while for acid treatment, optimum amylase production was measured as 0.2816 IU/mL/min as reported in Table 5. The linear and quadratic effects of the pretreatment conditions were observed using analyses of variance (ANOVA). The significance was revealed by Fisher’s F-test and probability *p*-value. In case of dilute acid treatment, F-test values of 33.93, 22.68, and 62.08 and *p*-values of 0.002, 0.002, and 0.001 were observed for amylase production, respectively. The results revealed that the substrate conc. (X1), time (X3), and squared coefficient of substrate conc. (X_1_^2^) were found to be the significant parameters during the dilute H_2_SO_4_ treatment for amylase production.

For the acid plus autoclaved treatment, the F-values of 43.70, 36.82, and 17.21 and *p*-values of 0.001, 0.002, and 0.009 were observed for amylase production, respectively. The results revealed that the X_1_, X_2_, X_1_^2^, X_2_^2^, X_3_^2^, X1*X2, and X1*X3 were found to be the significant parameters during the dilute H_2_SO_4_ assisted autoclaving treatment for amylase production, as reported in Table 6.

For cellulase production, dilute H_2_SO_4_ treatment was proven to be more effective and released up to 1.269 IU/mL/min with the experimental conditions of 10 g of PPW, 0.8% H_2_SO_4_ conc., and 4 h, as shown in Table 7. In the case of the dilute H_2_SO_4_ treatment, the *p*-values were observed to be 0.001, 0.002, and 0.003, whereas in the case of the acid assisted autoclaved treatment, the *p*-values were observed as 0.675, 0.703, and 0.770, respectively. The results revealed that X1, X3, X1*X1, X_2_^2^*X2 (%), X_3_^3^, and X2*X3 were found to be the significant parameters during the dilute H_2_SO_4_ treatment, while no such significance was observed in the acid assisted autoclaved treatment (Table 6).

### 3.6. Model Validation

The accuracy of the model was determined by the determination coefficient R^2^. The closer the value of R^2^ to 1 specifies a good correlation between the predicted and observed values. In the present study, the values of R^2^, R^2^ (pred), and R^2^ (adj) were observed as 0.93, 0.00, and 0.82, respectively, for amylase produced from the acid treated PPW. The values of R^2^, R^2^ (pred), and R^2^ (adj) were measured to be 0.96, 0.43, and 0.89, respectively, for amylase produced from the acid plus autoclaved PPW.

For cellulase production, the values of R^2^, R^2^ (pred.), and R^2^ (adj.) were observed as 0.98, 0.78 and 0.96, respectively, from the acid treated PPW. The values of R^2^, R^2^ (pred), and R^2^ (adj) were measured as 0.49, 0.00, and 0.00, respectively, for amylase produced from the acid plus autoclaved PPW.

### 3.7. Contour Plots for Amylase and Cellulase Production

The contour plots illustrate the interaction of different pretreatment conditions on the units of amylase and cellulase produced. In these plots, three main colors were revealed such as light green, blue, and dark green. The light green and blue signified the lowest yield while dark green signified the highest yield, as depicted in Figure 4. One study reported a high production of amylase using RSM and surface subplots [40]. The plots in this study reported s high amylase production with an increase in pH and less ammonium sulfate. Another report also reported high amylase production using 3D contour plots and depicted the significant effect of multiple variables [41].

For cellulase production, the contour plots illustrate the interaction of different process parameters on enzyme yield. In these plots, three main colors were observed such as light green, blue, and dark green. The light green color signified the lowest, the blue color signified moderate, while dark green color signified the highest yield, as shown in Figure 5. A previous research report showed high cellulase production using contour plots [42]. The report presented the effect of incubation time and malt extract. The contour plots revealed that high cellulase production was attained at an increased incubation time whereas less malt concentration favored more cellulase production. Research reports have also reported cellulase production using contour plots [43,44] and showed the interactive effects of multiple parameters on cellulase production.

### 3.8. Surface Plots for Amylase Production

The relative effects of different variables were studied by plotting 3D surface plots between two variables while keeping other factors at the zero level. Figure 6A–C represent the surface plots after dilute sulfuric acid treatment. Figure 6A displays the interactive effect of X1 and X2, and their combined effect favored more amylase production. Figure 6B depicts the combined effect of X1 and X3, where both of these parameters steadily favored more amylase production. Figure 6C depicts that both X2 and X3 did not significantly caused high amylase liberation. Figure 6D–F represent the surface plots after acid assisted autoclaved treatment and signify that the combined effect of all variables (i.e., X1 *X2, X1*X3, and X2*X3) significantly increased the concentration of amylase. These results are comparable to several previous reports. In fact, the increased liberation of amylase by optimizing different variables of fermentation has been well-documented [45,46,47,48].

### 3.9. Surface Plots for Cellulase Production

The effect of different process parameters and their interactions on the cellulase yield was described by 3D surface plots by plotting the response on the *Z*-axis against any two parameters while keeping the other factor constant (0 level). Figure 7A–C represent plots depicting the effect of dilute sulfuric acid treatment on the liberation of cellulase. Figure 7A reveals that the cellulase yield increased from 0.2 to 0.4 IU/mL when the substrate conc. and H_2_SO_4_ conc. increased. Figure 7C displays an almost similar trend where the increase in the value of acid conc. and time caused an increase in the cellulase yield. Figure 7B reports a high yield of cellulase, depicting that both the substrate conc. and time have the optimum effect on enzyme production. Figure 7D–F represent plots depicting the effect of dilute sulfuric acid treatment on the liberation of cellulase. All of these plots reported that a combined effect of all three parameters did not have significant effect on the response yield. An increase in the conc. of parameters favored the high production of the cellulase enzyme. The identification of specific conditions for the higher production of desired products adds to the economics of the process. Likewise, the optimization of cellulase yield using RSM has been reported [49]. Furthermore, high cellulase yields have been reported by describing the multiple effects of various parameters using 3D surface plots [50,51].

## 4. Conclusions

In the present work, the production of amylase and cellulase was optimized using acid treated PPW. The thermo-tolerant bacterium *Bacillus subtilis* QY4 (PP439596) was isolated from the local environment and screened for amylase and cellulase production. Different pretreatment techniques were investigated for their role in enzyme production. Acid assisted autoclaving treatment proved to be more effective for amylase production while dilute acidic treatment favored a greater production of cellulase. The novel bacterial strain *Bacillus subtilis* QY4 can be used for amylase and cellulase production because it could be considered a cost-effective source for these enzymes as it requires a relatively low incubation temperature to produce the amylase using acid treated PPW after three days of submerged fermentation. Our results strongly recommend high yield of enzymes that can be used for different industrial applications, particularly biofuel.

## Figures and Tables

**Figure 1 microorganisms-12-01106-f001:**
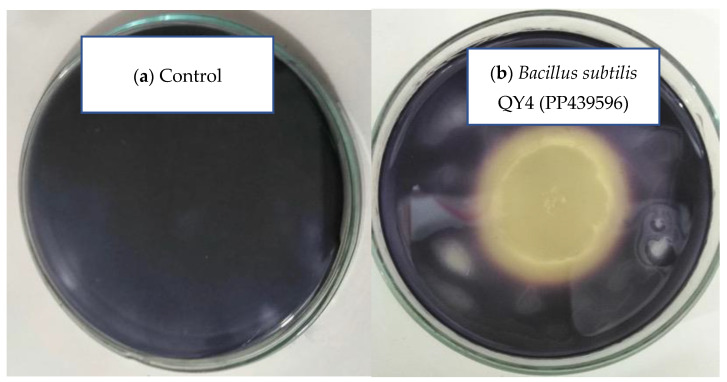
Effect of Gram’s iodine on the amylolytic zone in starch agar plates. (**a**) Control. (**b**) Inoculated with *Bacillus subtilis* QY4 (PP439596) with the hydrolysis zone indicating amylase yielding bacterium.

**Figure 2 microorganisms-12-01106-f002:**
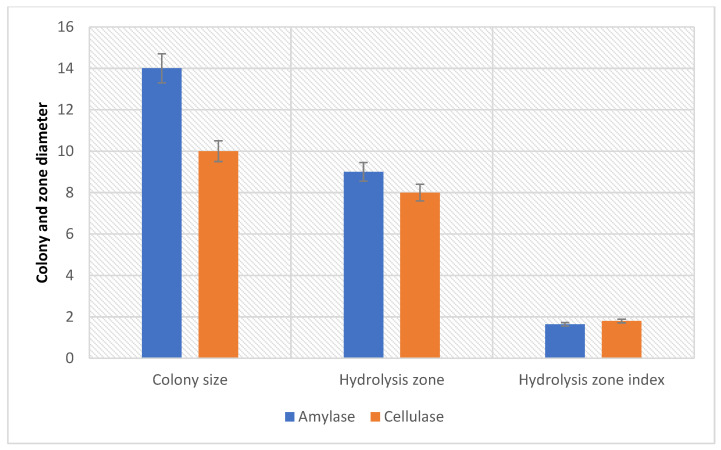
Amylolytic and cellulolytic potential of *B. subtilis* QY4 as envisaged by the bacterial colony sizes, hydrolytic zones, and hydrolytic indices on respective selective media. Hydrolysis zone index = colony diameter + hydrolysis zone diameter/colony diameter.

**Figure 3 microorganisms-12-01106-f003:**
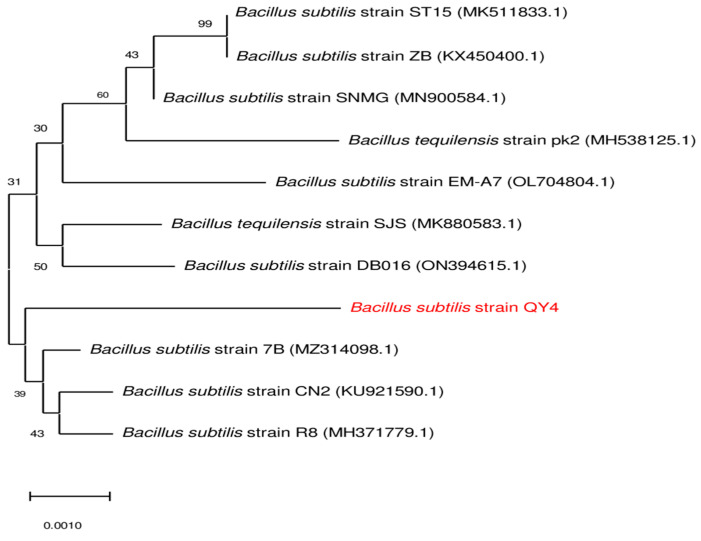
Phylogenetic tree of newly isolated *Bacillus subtilis* using the neighbor-joining method.

**Figure 4 microorganisms-12-01106-f004:**
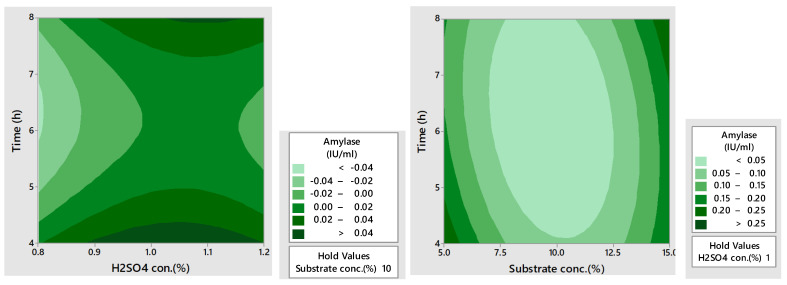
Contour plots for amylase production from the dilute acid and acid assisted autoclaved treated PPW.

**Figure 5 microorganisms-12-01106-f005:**
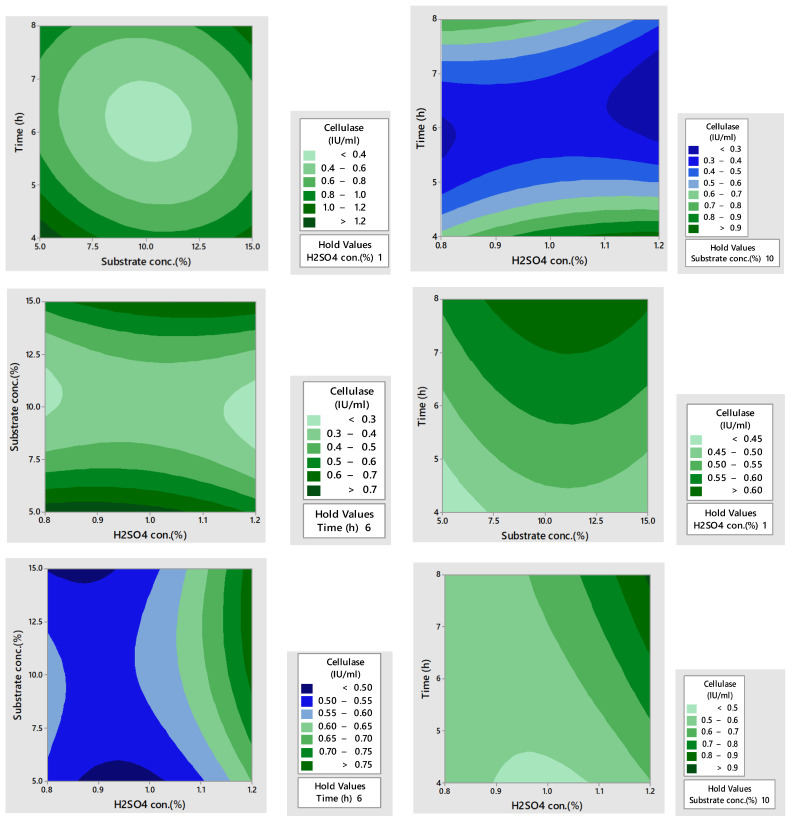
Contour plots for cellulase production from dilute acid and acid assisted autoclaved treated PPW.

**Figure 6 microorganisms-12-01106-f006:**
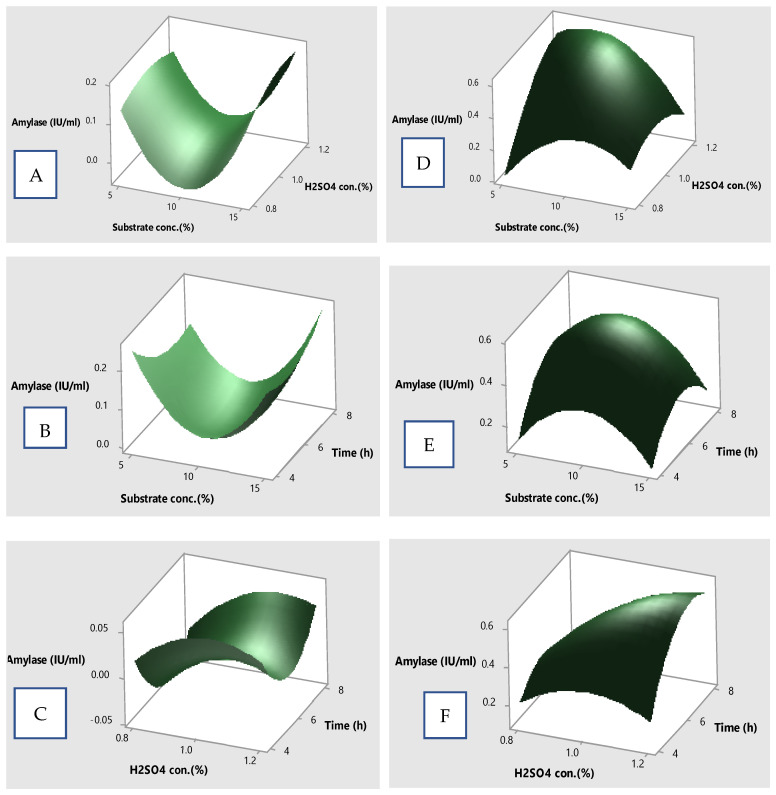
Surface plots showing the relative effect of different variables on amylase production; (**A**–**C**) representing acid assisted treatment, (**D**–**F**) representing acid assisted steam treatment.

**Figure 7 microorganisms-12-01106-f007:**
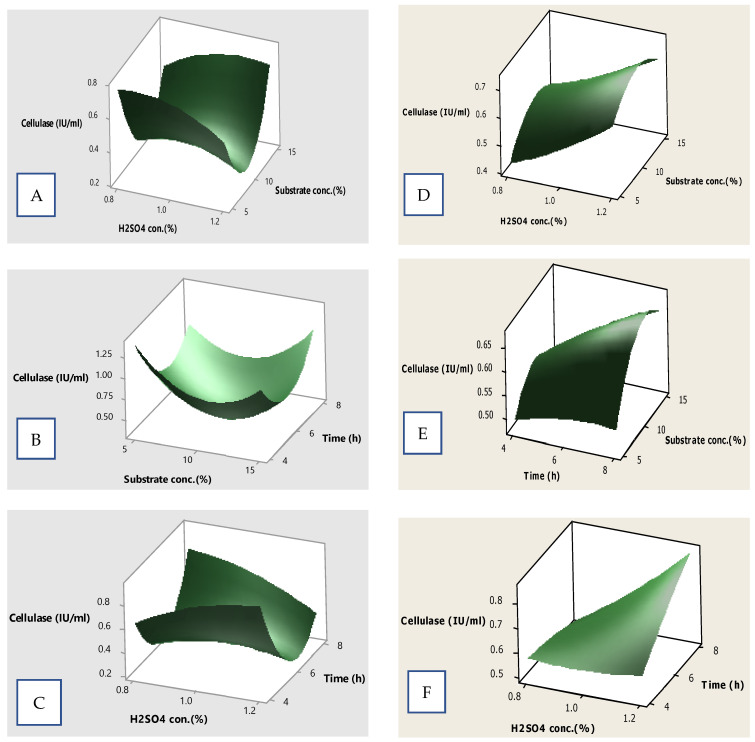
Surface plots showing the effect of different variables on cellulase yield; (**A**–**C**) representing acid assisted treatment, (**D**–**F**) representing acid assisted steam treatment.

**Table 1 microorganisms-12-01106-t001:** Composition of various culture media used in this study.

Culture Medium	Ingredients g/L	pH	Growth of Isolated Bacterium	Negative Control (*E. coli* ATCC 25922)
Nutrient agar	Peptone 5 g, yeast extract 1.5 g, beef extract 1.5 g, NaCl 5 g, agar 15 g	7.1–7.5	++	++
Nutrient broth	Peptone 5 g, meat extract 3 g, NaCl 5 g, agar 15 g	7.1–7.5	++	++
Commercial media for amylase and cellulase production	K_2_HPO_4_ 7 g, KH_2_PO_4_ 2 g, NaCl 5 g, MgSO_4_.7H_2_O 0.1 g (NH_4_)_2_SO_4_ 1 g, yeast extract 0.03 g, agar 13 g, starch for amylase/CMC for cellulose 5 g	7.0–7.5	++	− −
Self-constructed indigenous media	PPW 50 g and yeast extract 10 g	7.0–7.5	++	− −

++, good growth; − −, no growth.

**Table 2 microorganisms-12-01106-t002:** Codes and range of various parameters used for BBD.

Level				
Factor	Code	−1	0	1
Substrate conc. (%)	X_1_	5	10	15
H_2_SO_4_ conc. (%)	X_2_	0.8	1.0	1.2
Time (h)	X_3_	4	6	8

**Table 3 microorganisms-12-01106-t003:** Physicochemical characterization of potato peel waste.

Sr No.	Parameters	% Dry Weight PPW
1.	Moisture	10.29
2.	Ash	8.81
3.	Crude fat	0.44
4.	Crude protein	15.45
5.	Crude fiber	4.4
6.	Carbohydrates (mainly starch)	65.01

**Table 4 microorganisms-12-01106-t004:** Cultivation parameters of the isolated thermo-tolerant bacterium.

Temperature Range/Optimal	40–60 °C
pH range/optimal	6–7
Gram staining	Positive
Colony morphology	Creamy white, circular outline without edges, convex surface

**Table 5 microorganisms-12-01106-t005:** Results of BBD experiments on amylase and cellulase production in terms of IU/mL.

Run No.	X1	X2	X3	Amylase IU/mL Production after Acid Treatment		Amylase IU/mL Production after Acid Assisted Autoclaved Treatment		Cellulase IU/mL Production after Acid Treatment		Cellulase IU/mL Production after Acid Assisted Autoclaved Treatment	
				Obs.	Pred.	Obs.	Pred.	Obs.	Pred.	Obs.	Pred.
1	5	0.8	6	0.0885	0.1325	0.0432	0.0258	0.7898	0.826	0.624	0.547
2	5	1.2	6	0.1270	0.1209	0.1497	0.2061	0.3632	0.645	0.635	0.561
3	15	0.8	6	0.1261	0.1321	0.6010	0.5445	0.7719	0.587	0.514	0.590
4	15	1.2	6	0.2363	0.1921	0.1604	0.1777	0.6670	0.676	0.691	0.561
5	10	0.8	4	0.2816	0.2483	0.1238	0.1315	1.269	1.419	0.511	0.513
6	10	1.2	4	0.1822	0.1991	0.1704	0.1043	1.1911	1.049	0.507	0.560
7	10	0.8	8	0.1745	0.1576	0.2596	0.3257	0.8744	0.925	0.501	0.441
8	10	1.2	8	0.2220	0.2552	0.1742	0.1664	1.269	1.202	0.502	0.631
9	5	1.0	4	0.0290	0.0181	0.1960	0.2056	0.7375	0.717	0.374	0.440
10	15	1.0	4	0.0130	0.0401	0.1781	0.2268	0.8785	0.954	0.628	0.625
11	5	1.0	8	0.0145	−0.0126	0.1587	0.1099	0.8718	0.887	0.518	0.465
12	15	1.0	8	0.0255	0.0363	0.5885	0.5788	0.3380	0.444	1.116	0.989
13	10	1.0	6	0.0026	0.0019	0.5742	0.5567	0.3471	0.349	0.445	0.440
14	10	1.0	6	0.0014	0.0019	0.5640	0.5567	0.3410	0.349	0.540	0.613
15	10	1.0	6	0.0019	0.0019	0.5321	0.5567	0.3421	0.349	0.698	0.824

**Table 6 microorganisms-12-01106-t006:** ANOVA for amylase production.

Acidic Treatment					
Source	DF	Adj SS	Adj MS	F-Value	*p*-Value
Regression	9	0.126099	0.014011	8.29	0.016
Linear	3	0.070219	0.023406	13.85	0.007
X1	1	0.057323	0.057323	33.93	0.002
X2	1	0.001618	0.001618	0.96	0.373
X3	1	0.008578	0.008578	5.08	0.074
Square	3	0.114948	0.038316	22.68	0.002
X1*X1	1	0.104885	0.104885	62.08	0.001
X2*X2	1	0.002502	0.002502	1.48	0.278
X3*X3	1	0.007334	0.007334	4.34	0.092
Interaction	3	0.006862	0.002287	1.35	0.357
X1*X2	1	0.001285	0.001285	0.76	0.423
X1*X3	1	0.005395	0.005395	3.19	0.134
X2*X3	1	0.000182	0.000182	0.11	0.756
Residual Error	5	0.008448	0.001690		
Lack-of-Fit	3	0.008447	0.002816	7749.78	0.000
Pure Error	2	0.000001	0.000000		
Total	14	0.134547			
**Acid Assisted Autoclaving Treatment**					
**Source**	**DF**	**Adj SS**	**Adj MS**	**F-Value**	***p*-Value**
Regression	9	0.569760	0.063307	14.56	0.004
Linear	3	0.212159	0.070720	16.26	0.005
X1	1	0.190047	0.190047	43.70	0.001
X2	1	0.044152	0.044152	10.15	0.024
X3	1	0.018940	0.018940	4.35	0.091
Square	3	0.270078	0.090026	20.70	0.003
X1*X1	1	0.160141	0.160141	36.82	0.002
X2*X2	1	0.044623	0.044623	10.26	0.024
X3*X3	1	0.102369	0.102369	23.54	0.005
Interaction	3	0.129294	0.043098	9.91	0.015
X1*X2	1	0.074830	0.074830	17.21	0.009
X1*X3	1	0.004356	0.004356	1.00	0.363
X2*X3	1	0.050109	0.050109	11.52	0.019
Residual Error	5	0.021746	0.004349		
Lack-of-Fit	3	0.020781	0.006927	14.36	0.066
Pure Error	2	0.000965	0.000482		
Total	14	0.591506			

**Table 7 microorganisms-12-01106-t007:** ANOVA for cellulase production.

Acidic Treatment					
Source	DF	Adj SS	Adj MS	F-Value	*p*-Value
Regression	9	1.45681	0.16868	39.31	0.000
Linear	3	0.70942	0.236473	57.42	0.000
X1	1	0.41226	0.412262	100.11	0.000
X2	1	0.01350	0.013495	3.28	0.130
X3	1	0.27248	0.272482	66.17	0.000
Square	3	1.13487	0.378292	91.86	0.000
X1*X1	1	0.49535	0.495350	120.29	0.000
X2*X2	1	0.00374	0.003740	0.91	0.384
X3*X3	1	0.69363	0.693627	168.43	0.000
Interaction	3	0.23847	0.079489	19.30	0.004
X1*X2	1	0.01823	0.018225	4.43	0.089
X1*X3	1	0.10485	0.104846	25.46	0.004
X2*X3	1	0.11540	0.115396	28.02	0.003
Residual Error	5	0.02059	0.004118		
Lack-of-Fit	3	0.02022	0.006740	36.40	0.027
Pure Error	2	0.00037	0.000185		
Total	14				
**Acid Assisted Autoclaving Treatment**					
**Source**	**DF**	**Adj SS**	**Adj MS**	**F-Value**	***p*-Value**
Regression	9	0.154048	0.017116	0.55	0.796
Linear	3	0.095666	0.006321	0.20	0.891
X1	1	0.006161	0.006184	0.20	0.675
X2	1	0.070688	0.002988	0.10	0.770
X3	1	0.018818	0.008146	0.26	0.632
Square	3	0.024076	0.022010	0.26	0.854
X1*X1	1	0.022423	0.000980	0.70	0.440
X2*X2	1	0.001103	0.000550	0.03	0.866
X3*X3	1	0.000550	0.034305	0.02	0.900
Interaction	3	0.034305	0.000462	0.37	0.781
X1*X2	1	0.000462	0.005112	0.01	0.908
X1*X3	1	0.005112	0.028730	0.16	0.703
X2*X3	1	0.028730	0.156364	0.92	0.382
Residual Error	5	0.156364	0.082395		
Lack-of-Fit	3	0.082395	0.073969	0.74	0.617
Pure Error	2	0.073969			
Total	14	0.310411			

## Data Availability

Sequencing data from this study were deposited at NCBI GenBank with accession number PP439596. Meanwhile, the strain has been deposited in the Microbial Stock at the Microbial Biotechnology Laboratory located at the Institute of Zoology, University of Punjab, Lahore, Pakistan and Microbial Stock Conservation at the Department of Research and Development at Paktex Industries, Kamoke, Gujranwala, Pakistan, with conservation numbers MBLZPU-9010 and R&D/PI202-1, respectively.

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
