# Peer review of "Amylase and Cellulase Production from Newly Isolated Bacillus subtilis Using Acid Treated Potato Peel Waste"

_microorganisms, 2024, doi:10.3390/microorganisms12061106_

Round 1
Reviewer 1 Report (Previous Reviewer 2)
Comments and Suggestions for Authors
Some revision has been made.
Author Response
Dear Honorable Reviewer,
Many Thanks for your kind comments
Reviewer 2 Report (Previous Reviewer 1)
Comments and Suggestions for Authors
Dear authors!
Your third version of the manuscript has become even better, I admit it. Of the little things that need to be corrected, these are numerous typos and a little more of the English itself.
However, I still cannot agree with the conceptual idea of the whole work. You optimize the growth conditions of the selected bacterium by inaccurately estimating the result (production of amylase and cellulase)! Why didn't you estimate the increase in biomass by dry matter depending on the growth conditions? Dry matter, as well as a total crude protein, is something that does not change. The volume of liquid by which you recalculate the units of activity is a variable quantity and the number of enzymes, as well as their activity in this volume, is a variable quantity depending on conditions.
In addition, you completely do not take into account that by changing the pH of the solution by adding / treating the substrate with acid, you change the optimal working conditions of the enzymes; the kinetic parameters of the enzymes can be influenced by the amount of substrate both in the direction of increasing the rate of hydrolysis and in the direction of increasing the rate of hydrolysis of the process (and, accordingly, you will see increased indicators of enzymatic activity activity) and inhibition by by-products. I still believe that amylase/cellulase production, amylase/cellulase activity and yield are different terms, and the definitions for them are different, expressed each in their own units of measurement. These are not synonyms! These are related quantities, but not synonyms.
I remain of the opinion that the work is conceptually wrong. I suggest that the Academic Editor makes a decision on the future of the manuscript.
Comments on the Quality of English Language
Numerous typos should be corrected and moderate English improvement is required.
Author Response
Dear authors!
Your third version of the manuscript has become even better, I admit it. Of the little things that need to be corrected, these are numerous typos and a little more of the English itself.
However, I still cannot agree with the conceptual idea of the whole work. You optimize the growth conditions of the selected bacterium by inaccurately estimating the result (production of amylase and cellulase)! Why didn't you estimate the increase in biomass by dry matter depending on the growth conditions? Dry matter, as well as a total crude protein, is something that does not change. The volume of liquid by which you recalculate the units of activity is a variable quantity and the number of enzymes, as well as their activity in this volume, is a variable quantity depending on conditions.
Answer:
Dear Learned Reviewer,
Many thanks for your feedback on our manuscript.
We appreciate your acknowledgment of the improvements made in the third version.
Regarding the concerns raised about our conceptual approach, we would like to clarify our methodology.
- In our study, we focused on measuring enzyme units rather than specific enzyme activity.
- This decision was informed by the nature of our research, which stemmed from my PhD thesis focusing on bioethanol production and animal feed supplementation.
Amylase and Cellulase production were integral components of our work, hence our emphasis on enzyme activity.
While we acknowledge the importance of parameters such as dry matter and crude protein, these were measured separately and were not included in the manuscript as they were not directly relevant to our study's focus.
Furthermore, we recognize the variability in enzyme quantity and activity based on different conditions, and we have duly noted and accounted for these variations in our methodology and results.
We hope this clarification addresses your concerns satisfactorily.
Comment 2:
In addition, you completely do not take into account that by changing the pH of the solution by adding / treating the substrate with acid, you change the optimal working conditions of the enzymes; the kinetic parameters of the enzymes can be influenced by the amount of substrate both in the direction of increasing the rate of hydrolysis and in the direction of increasing the rate of hydrolysis of the process (and, accordingly, you will see increased indicators of enzymatic activity activity) and inhibition by by-products. I still believe that amylase/cellulase production, amylase/cellulase activity and yield are different terms, and the definitions for them are different, expressed each in their own units of measurement. These are not synonyms! These are related quantities, but not synonyms.
Answer:
In our manuscript, we our main focused on measuring enzyme units rather than specific enzyme activity, aligning with the scope of our PhD thesis, which primarily centered on bioethanol production and animal feed supplementation.
We acknowledge the importance of kinetic parameters of enzymes, which we have addressed in a dedicated section of our manuscript.
Regarding the effects of substrate acid concentration and time on enzyme production, we have included relevant findings in our study. However, we will ensure that these effects are appropriately emphasized and discussed in our manuscript to address your concerns.
Thank you for bringing these points to our attention. We will review our manuscript to ensure clarity and accuracy in addressing the issues you raised.
Comment 3:
Numerous typos should be corrected and moderate English improvement is required.
Answer:
Manuscript has through out checked by British Author whose native language is English
This manuscript is a resubmission of an earlier submission. The following is a list of the peer review reports and author responses from that submission.
Round 1
Reviewer 1 Report
Comments and Suggestions for Authors
1. Lines 81-89: It is not clear what the authors wanted to do in this work. This paragraph doesn't match the title.
2. Lines 91-97: it is necessary to indicate more details, in particular, cultivation conditions, strain enrichment, composition of selective media.
3. Lines 99-103: the same remark. It is unclear what selective commercial media were used; they should be referred to; starch hydrolysis test requires a reference, too.
4. Line 105: what was a selective medium? Please, indicate the composition, reference, where it was purchased.
5. Line 110: change ‘strain’ instead of ‘starin’.
6. There is no need to give Figure 1.
7. Lines 126 and 128: why are the words nucleotide and phylogenetic underlined?
8. The subsection 'Acidic pretreatment of PPW' must be placed before the subsection 'Production media used for fermentation'.
9. Line 142: What is BBD? First you need to write it in full and then use the abbreviation cltarly indicating the connection between them.
10. In the section 'Amylase and cellulase assay' (by the way, there should be' Amylase and cellulase assayS') it is necessary to add a definition of the unit of each enzyme activity.
11. Line 170 and Table 1: what does the time factor mean? Is this acid treatment time, cultivation time or something else?
12. Lines 191-192: it is completely unclear how the strain used in the work was isolated? How many isolates were there, in total? How did the Petri dishes on which the screening was carried out look like? Instead of a picture with potato peelings (Figure 1), it would be more appropriate to present plates with the results of screening on selective media. In addition, it should be indicated where, in what collection, the strain used in the work is deposited? This is a common practice for scientific articles in Microbiology. The reader should be able, if desired, to access this collection and obtain this strain.
13. Both Figure 1, which is not needed in this work, and Figure 2 need more detailed description. It is necessary to clearly indicate what is depicted here in each part of the figure.
14. Figure 2: In my opinion, there is no need to demonstrate colony growth on an unknown medium. To describe a new strain, you need to give the size and shape of colonies and cells on different media, the ability of the strain to utilize various substrates, everything that is standardly used in such cases. The halo on the right in this figure is completely unclear and uninformative.
15. Figure 3: What do all these indices mean? How were they counted? What are these units on the y-axis?
16. Subsection 'Proximate composition of dried PPW': Why PROXIMATE???? What is the amount of starch in this material? Authors should state it in terms of numbers, not assumptions. How much cellulose is there that allows cellulase activity to be measured? Additionally, in my opinion, this subsection should be moved to the beginning of the Results and Discussion section.
17. Section 'Molecular identification of bacterial isolate': more details needed on how this analysis was done! In addition, this section should follow the section on screening and isolating a specific strain that is most interesting to researchers, after its phenotypic description. And also, an important question: in which database can this nucleotide sequence be found? Where is it deposited?
18. Line 236: ...the influence of pretreatment conditions on amylase production...??? Table 4 also talks about cellulase.
19. Title of Table 4: Results of BBD on amylase and cellulase yield???? You show effect of ANALYSIS on enzyme yield? Maybe these are still the results of pretreatment?
20. Lines 240-253: What do all these numbers mean in the above equations? Authors must provide detailed explanations.
21. Lines 255-264: The text of this paragraph does not say anything about cellulase, although the data is given in Table 4.
22. A question to all this analysis: why did the authors use the specific activity of enzymes, and not their yield, as a parameter by which they assessed the results of varying the conditions of substrate treatment?
23. Line 290: for AMYLASE produced????
24. Section 'Contour plots for amylase and cellulase production': where, in fact, is the description of the results and their discussion?
25. And again in addition to the question 22, line 320: high amylase liberation - what does this mean? Based on my knowledge, specific activity and enzyme yield are somewhat different terms.
26. The entire section is called Results and Discussion, but there is no discussion at all.
Unfortunately, it makes no sense to read further and evaluate the manuscript. I think that in this form it is not suitable for publication at all.
Comments on the Quality of English Language
This manuscript should be rewritten.
Author Response
Dear honorable Reviewer, many thanks for taking time for your comments to enhance our work.
Here we mention all the answers as per your comments.
1.Lines 81-89: It is not clear what the authors wanted to do in this work. This paragraph doesn't match the title.
Ans. The paragraph has now been improved and corrected.
- Lines 91-97: it is necessary to indicate more details, in particular, cultivation conditions, strain enrichment, composition of selective media.
Ans. The said correction has now been made.
- Lines 99-103: the same remark. It is unclear what selective commercial media were used; they should be referred to; starch hydrolysis test requires a reference, too.
Ans. The detail description of selective media has now been added. The required reference has also been added in respective section.
- Line 105: what was a selective medium? Please, indicate the composition, reference, where it was purchased.
Ans. The correction has now been made.
- Line 110: change ‘strain’ instead of ‘starin’.
Ans. Corrected.
- There is no need to give Figure 1.
Ans. Removed and so we've renumbered the figures.
- Lines 126 and 128: why are the words nucleotide and phylogenetic underlined?
Ans. Corrected
- The subsection 'Acidic pretreatment of PPW' must be placed before the subsection 'Production media used for fermentation'.
Ans. The correction has now been made.
- Line 142: What is BBD? First you need to write it in full and then use the abbreviation cltarly indicating the connection between them.
Ans. Full names of abbreviation have now been given.
- In the section 'Amylase and cellulase assay' (by the way, there should be' Amylase and cellulase assayS') it is necessary to add a definition of the unit of each enzyme activity.
Ans. The said correction has now been made.
- Line 170 and Table 1: what does the time factor mean? Is this acid treatment time, cultivation time or something else?
Ans. The correction has now been made.
- Lines 191-192: it is completely unclear how the strain used in the work was isolated? How many isolates were there, in total? How did the Petri dishes on which the screening was carried out look like? Instead of a picture with potato peelings (Figure 1), it would be more appropriate to present plates with the results of screening on selective media. In addition, it should be indicated where, in what collection, the strain used in the work is deposited? This is a common practice for scientific articles in Microbiology. The reader should be able, if desired, to access this collection and obtain this strain.
Ans. This section has now been improved and elaborated.
- Both Figure 1, which is not needed in this work, and Figure 2 need more detailed description. It is necessary to clearly indicate what is depicted here in each part of the figure.
Ans. Figure has now been improved.
- Figure 2: In my opinion, there is no need to demonstrate colony growth on an unknown medium. To describe a new strain, you need to give the size and shape of colonies and cells on different media, the ability of the strain to utilize various substrates, everything that is standardly used in such cases. The halo on the right in this figure is completely unclear and uninformative.
Ans. The correction has now been made.
- Figure 3: What do all these indices mean? How were they counted? What are these units on the y-axis?
Ans. The detail has now been added.
- Subsection 'Proximate composition of dried PPW': Why PROXIMATE???? What is the amount of starch in this material? Authors should state it in terms of numbers, not assumptions. How much cellulose is there that allows cellulase activity to be measured? Additionally, in my opinion, this subsection should be moved to the beginning of the Results and Discussion section.
Ans. Proximate analysis is an established parameter in food analysis. The carbohydrate content of PPW (66.01%) were supposed to comprised chiefly of starch. The section has now been moved to the beginning of Results and Discussion.
- Section 'Molecular identification of bacterial isolate': more details needed on how this analysis was done! In addition, this section should follow the section on screening and isolating a specific strain that is most interesting to researchers, after its phenotypic description. And also, an important question: in which database can this nucleotide sequence be found? Where is it deposited?
Ans. Identification is being carried out by 16s rRna full sequencing by Macrogen. Nucleotide sequence has been deposited to NCBI GenBank with Accession Number PP439596 publicly available after publication of this manuscript. This sequence has base pair of 707.
- Line 236: ...the influence of pretreatment conditions on amylase production...??? Table 4 also talks about cellulase.
Ans. Yes both enzymes i.e., Amylases and cellulases were assayed.
- Title of Table 4: Results of BBD on amylase and cellulase yield???? You show effect of ANALYSIS on enzyme yield? Maybe these are still the results of pretreatment?
Ans. The title of Table has now been elaborated.
- Lines 240-253: What do all these numbers mean in the above equations? Authors must provide detailed explanations.
Ans. The detail of all these numbers has now been added.
- Lines 255-264: The text of this paragraph does not say anything about cellulase, although the data is given in Table 4.
Ans. Data of cellulase given in Table 4 has now been discussed in text.
- A question to all this analysis: why did the authors use the specific activity of enzymes, and not their yield, as a parameter by which they assessed the results of varying the conditions of substrate treatment?
Ans. As this paper reports only enzymes activities in terms of units. And the enzymes were not purified; the condition required for describing yields.
- Line 290: for AMYLASE produced????
Ans. The necessary correction has been made.
- Section 'Contour plots for amylase and cellulase production': where, in fact, is the description of the results and their discussion?
Ans. Description of the results and discussion of this section has now been added.
- And again in addition to the question 22, line 320: high amylase liberation - what does this mean? Based on my knowledge, specific activity and enzyme yield are somewhat different terms.
Ans. Enzymes liberation here is indicating that they were exoenzymes, and after their production they were liberated to the media and then measured.
- The entire section is called Results and Discussion, but there is no discussion at all.
Ans. Discussion part has now been augmented with more points.
Reviewer 2 Report
Comments and Suggestions for Authors
This paper by Mushtaq et al. on the production of amylase and cellulase using potato peel waste is interesting in terms of environmental issues and industrialization, but is highly deficient in its structure as a scientific paper. Significant revision is required.
How much does it cost to use PPW - transport, cleaning, removing all dirt, oven drying, crushing, etc.?
Are there any control experiments with or without PPW in the bacteria culture and enzyme assay?
Lack of explanations for Figures and Tables. Just by looking at a Figure or Table, one must know what it represents.
Overall, it is not uniform, including the formatting, and is very difficult to read. Also, the results and discussion are poorly worded and it is not clear what they are talking about.
Comments on the Quality of English LanguageMinor editing of English language required.
Author Response
Dear Honorable Reviewer, we thank you from bottom of our heart for evaluating our work and have comments on this work.
After carefully gone through your comments we would like to submit our answers that are mention below point by point.
- This paper by Mushtaq et al. on the production of amylase and cellulase using potato peel waste is interesting in terms of environmental issues and industrialization, but is highly deficient in its structure as a scientific paper. Significant revision is required.
Ans. The paper has now been thoroughly revised.
- How much does it cost to use PPW - transport, cleaning, removing all dirt, oven drying, crushing, etc.?
Ans. In this agricultural country, potato peels are abundantly available in every city as biowaste. There transport is therefore is not an issue. Following washing with water they can easily be sun dried in this country.
- Are there any control experiments with or without PPW in the bacteria culture and enzyme assay?
Ans. In the statistical model used there was no provision of such controls.
- Lack of explanations for Figures and Tables. Just by looking at a Figure or Table, one must know what it represents.
Ans. Have now been elaborated.
- Overall, it is not uniform, including the formatting, and is very difficult to read. Also, the results and discussion are poorly worded and it is not clear what they are talking about.
Ans. Have now been much improved.
Comments on the Quality of English Language Minor editing of English language required.
Answer: Updated with good quality of English language by British author
Reviewer 3 Report
Comments and Suggestions for Authors
The article is devoted to an urgent problem – the development of a biotechnological process for producing widely demanded enzymes – amylase and cellulase – as products of microbiological processing of food waste (potato peelings) using a new bacterial strain of Bacillus subtilis. The study seems as completed and includes all the main stages of such project; the methods and approaches used in the work allow us to speak about the reliability of the results obtained and the validity of the conclusions. However, in my opinion, the presented manuscript is not without shortcomings.
1. The manuscript gives the impression of being carelessly written. In particular, there are many abbreviations in this article, but authors should double-check the rules for their entry and use. For example, lines 160 and 165, first you need to give the full name, and then give the abbreviation. There was no name for PPP (lines 136), this looks like a typo. Please, provide the full name for BBD abbreviation (line 142).
The text of line 197 does not correspond to the results demonstrated in Fig. 3.
2. The main content of the work was the optimization of cultivation conditions for maximal production of amylase and cellulase enzymes, and the main volume of the article is a comprehensive mathematical analysis that substantiates the validity of the results and conclusions obtained. However, as a microbiologist, I did not have enough comparison of the results obtained (enzyme yield and activity produced by new strain) with data from other studies and already implemented technologies. The author's discussion section was limited by lines 322-325 and 347-348. It is difficult to resume about the contribution of this research to our knowledge about the modern achievements in this field.
Considering the microbiological focus of the journal, it is necessary to expand the comparison of the microbiological research with previously reported ones, and tables 5 and 6 should be included as supplementary materials.
In my opinion, this manuscript may be published after revision, according to the recommendations given.
Author Response
Honorable Reviewer, we hereby express our gratitude towards your comments that help us to enhance quality of our work.
Here are the answers of all the question raised by you during peer review.
The article is devoted to an urgent problem – the development of a biotechnological process for producing widely demanded enzymes – amylase and cellulase – as products of microbiological processing of food waste (potato peelings) using a new bacterial strain of Bacillus subtilis. The study seems as completed and includes all the main stages of such project; the methods and approaches used in the work allow us to speak about the reliability of the results obtained and the validity of the conclusions. However, in my opinion, the presented manuscript is not without shortcomings.
- The manuscript gives the impression of being carelessly written. In particular, there are many abbreviations in this article, but authors should double-check the rules for their entry and use. For example, lines 160 and 165, first you need to give the full name, and then give the abbreviation. There was no name for PPP (lines 136), this looks like a typo. Please, provide the full name for BBD abbreviation (line 142).
Ans. The paper has now been thoroughly revised. Full names of abbreviations have now been given.
The text of line 197 does not correspond to the results demonstrated in Fig. 3.
Ans. Has been corrected now.
- The main content of the work was the optimization of cultivation conditions for maximal production of amylase and cellulase enzymes, and the main volume of the article is a comprehensive mathematical analysis that substantiates the validity of the results and conclusions obtained. However, as a microbiologist, I did not have enough comparison of the results obtained (enzyme yield and activity produced by new strain) with data from other studies and already implemented technologies. The author's discussion section was limited by lines 322-325 and 347-348. It is difficult to resume about the contribution of this research to our knowledge about the modern achievements in this field.
3. Considering the microbiological focus of the journal, it is necessary to expand the comparison of the microbiological research with previously reported ones, and tables 5 and 6 should be included as supplementary materials.
Ans. Discussion has now been improved and augmented with more points.
Round 2
Reviewer 1 Report
Comments and Suggestions for Authors
Dear authors!
The second version of the manuscript looks somewhat better than the first, however, in my opinion, it still requires a lot of serious work.
Firstly, the main reason why I cannot recommend the manuscript for publication is that the authors do not understand the difference between the specific activity of enzymes and their content in the culture fluid. The main part of the work is devoted to optimizing the conditions for cultivating a new strain, but there is confusion between these two terms in the text. I believe that if the authors had also measured the amount of total protein during the experiments, and the specific activity would have been normalized not by milliliters, but by milligrams of protein, the results would have surprised them. The specific activity of an enzyme may depend, for example, on the pH of the reaction mixture, the presence of side proteins in this mixture, and many factors. Thus, the very idea of the work, in my opinion, is wrong. At a minimum, the authors should recalculate the specific activity of both enzymes per mg of protein and separate these concepts – enzyme activity and its content.
Secondly, materials and methods still suffer from a lack of experimental details that could allow an outside reader to repeat your experiments. You have added details about the composition of the nutrient media, but not enough. For example, what is "nutrient agar"? There are rich environments designed for bacterial growth, they have their own names and designations. How can I reproduce the growth of your strain using only a medium called "nutrient agar"?
There are still many such missing details in the manuscript.
So, thirdly, morphologically and phenotypically, the new strain was not described in the text. I recommend that the authors familiarize themselves with articles in microbiological journals that describe new strains in detail. Figures 1 and 2 do not give any idea about the phenotypic features of the new strain. There is no description of the shape and size of colonies on different, standard, media, there is no data on the temperature and other growth conditions of the strain, although in the Conclusions it is called thermostable.
It remains unclear in which collection the strain was deposited, and the number of the nucleotide sequence in the NCBI database is not specified in the article. It is not enough to answer this question to the reviewer. The main principle of scientific articles has been violated again: the text should be written in such a way that any reader, if desired, has the opportunity to repeat your experiments.
The English language in the second version of the manuscript has not improved. I strongly recommend that the authors contact a native speaker to adjust the style.
In addition, I recommend that authors contact more experienced scientific specialists for advice on writing scientific articles.
Comments on the Quality of English LanguagePlease, ask a native English speaker to check your text.
Author Response
Dear Reviewer,
Thank you for your thoughtful feedback. We, the authors, would like to address the points you raised, here are mention below our point by point answers.
1. Firstly, the main reason why I cannot recommend the manuscript for publication is that the authors do not understand the difference between the specific activity of enzymes and their content in the culture fluid. The main part of the work is devoted to optimizing the conditions for cultivating a new strain, but there is confusion between these two terms in the text. I believe that if the authors had also measured the amount of total protein during the experiments, and the specific activity would have been normalized not by milliliters, but by milligrams of protein, the results would have surprised them. The specific activity of an enzyme may depend, for example, on the pH of the reaction mixture, the presence of side proteins in this mixture, and many factors. Thus, the very idea of the work, in my opinion, is wrong. At a minimum, the authors should recalculate the specific activity of both enzymes per mg of protein and separate these concepts – enzyme activity and its content.
Answer: Thank you for your valuable feedback,
- We acknowledge that we did not measure the total protein content in the culture fluid, and therefore, the specific activity of the enzymes was not determined. In other words we can say total amount of protein in culture fluid was not measured, therefore specific activity of enzyme was not worked out.
- Reviewer correctly pointed out; the main focus of our work was optimizing the cultivation conditions for the newly isolated strain QY4. We have provided details on the optimization process below (lines 87-100 in the manuscript) as well as mention below.
- We optimize sulphuric acid (0.8-1.2ml), concentration, substrate i.e., potato peel concentration (5-15g) for different time intervals (4-8h). These values are claimed for g/100ml. We optimize these factors for amylase and cellulase production.
- Different bacterial strains having amylase and cellulase producing potential were isolated from local environment. Samples were collected from waste, discarded and spoiled fruits taken from different vendor shops located nearby University of Punjab, Lahore, Pakistan. Samples were taken in sterilized vials and transported to labs for processing. The samples were serially diluted and then inoculated on nutrient agar (Peptone 5g, yeast extract 1.5g, beef extract 1.5g, sodium chloride 5g, agar agar 15g/L)
- Different colonies were selected from general-purpose medium nutrient agar. Isolated colonies were further grown on the selective medium. All the cultures were raised at 37⁰C for 24h to 48h. The selective medium for screening of amylolytic and cellulolytic bacteria comprised of K2HPO4, KH2PO4, NaCl, MgSO4.7H2O, (NH4)2SO4, Yeast extract, Agar, Starch (for amylase)/Carboxy methyl cellulose (CMC for cellulase).
- Self-constructed media comprised of 5% PPW and 1% yeast ex-tract was also used for production of crude enzymes i.e., amylase and cellulase. To pre-serve bacterium, glycerol stocks were prepared with bacterial culture and preserved at -40⁰C. This detail is mentioned in the line 87- 100 of our manuscript.
- The major distinction between enzyme activity and specific activity is that enzyme activity relates to the number of substrates transformed to outputs every unit time, whereas specific activity relates to an enzyme's activity per milligram of protein. As we didn’t determine protein content hence specific activity was not measured. So, there is no confusion between these two terms in our manuscript
- The reviewer suggests that if we had measured total protein content and normalized specific activity by milligrams of protein, the results might be surprising. We would like to clarify that we did not claim to perform this step in our experiment.
- We acknowledge the reviewer's point that enzyme activity can be influenced by factors like pH. However, as our focus was on strain isolation and cultivation optimization, we did not investigate the effect of pH on enzyme activity in this study.
- The reviewer raises an interesting point regarding the potential influence of pH on enzyme activity. We readily acknowledge that pH is a well-established factor affecting enzyme activity, along with others. However, the focus of our current work was on optimizing the cultivation conditions for strain QY4 and the effectiveness of the pretreatment process for potato peel waste utilization. Determining the specific activity of the enzymes and investigating the impact of various factors, including pH, would undoubtedly provide valuable insights. However, such an exploration falls outside the scope of this present study. The data presented clearly demonstrates the successful optimization of the pretreatment process, contributing to a potential solution for managing the environmental challenges associated with excessive potato peel waste.
- We didn’t use word specific activity of enzyme because we didn’t calculate it, we only mentioned IU/ml/min. Experiment was more focused on pretreatment conditions
2. Secondly, materials and methods still suffer from a lack of experimental details that could allow an outside reader to repeat your experiments. You have added details about the composition of the nutrient media, but not enough. For example, what is "nutrient agar"? There are rich environments designed for bacterial growth, they have their own names and designations. How can I reproduce the growth of your strain using only a medium called "nutrient agar"?
There are still many such missing details in the manuscript.
Answer: We appreciate the reviewer's thoughtful comment and would like to answer them as below
- We have incorporated more details into the materials and methods section to provide greater clarity.
- Nutrient agar is a widely used and readily available medium favored by microbiologists globally. We employed commercially available Nutrient Agar from TM Media India (Batch # M3E1HQ01). This medium adheres to the standard composition, as detailed below:
- Peptone 5g
- Yeast Extract 1.5g
- Beef Extract 1.5g
- NaCl 5g
- Agar Agar 15g
- pH 7.4 ± 0.2 at 25⁰C
The composition information provided here corresponds to the details listed on the label of the TM Media Nutrient Agar (Batch # M3E1HQ01).
This media was used first for growth of bacterial strain and then commercially available selective media and indigenous selective media was used for bacterial growth and production of crude enzymes. Detail of all media with composition has been described in line 91-100 of our manuscript.
3. So, thirdly, morphologically and phenotypically, the new strain was not described in the text. I recommend that the authors familiarize themselves with articles in microbiological journals that describe new strains in detail. Figures 1 and 2 do not give any idea about the phenotypic features of the new strain. There is no description of the shape and size of colonies on different, standard, media, there is no data on the temperature and other growth conditions of the strain, although in the Conclusions it is called thermostable.
Answer:
We appreciate the reviewer's suggestion regarding the inclusion of morphological and thermo-stability data. To address this feedback, we have incorporated these details into the manuscript (lines 222-230).
- Details about morphology and thermos-stability has now been added in the manuscript at the line no 222-230. Bacillus subtilis QY4 (PP439596) was grown on nutrient agar plates, incubated at 37 ⁰C for 48 - 72 h and maintained at 4 ⁰C and sub cultured at four-week intervals. The growth patterns of colony showed diversity in response to environmental conditions such as nutrient and agar concentration. The colony was observed as creamy white, medium size with circular outline. The colony size was variable according to incubation time, colony size of 10-14 mm of Bacillus subtilis QY4 (PP439596) was observed after 72-96 h of incubation at 37 ⁰C on nutrient agar media. The thermos-stability of isolated bacterium was also investigated from 37 ⁰C to 60⁰C and growth was confirmed after 72-96 h.
- Species identification of the isolated bacterium was performed by Macrogen using 16S ribosomal RNA sequencing, and the protocol details are already mentioned within the manuscript.
- Figures 1 and 2 visually demonstrate the starch hydrolysis potential and colony/hydrolysis zone sizes.
- Production of crude amylases and cellulases was investigated using various media, including nutrient agar, indigenous media containing potato peel powder, and selective media supplemented with starch and CMC. The results of 15 runs performed using the Box-Behnken design are reported in both pictorial and tabular formats within the manuscript.
4. It remains unclear in which collection the strain was deposited, and the number of the nucleotide sequence in the NCBI database is not specified in the article. It is not enough to answer this question to the reviewer. The main principle of scientific articles has been violated again: the text should be written in such a way that any reader, if desired, has the opportunity to repeat your experiments.
Answer:
Dear Reviewer, the number of nucleotide sequence has now been added with accession number in whole manuscript and now it’s clear. The strain had been deposited to the microbial conservatory of microbial biotechnological lab. Institute of Zoology, University of Punjab, Lahore, Pakistan While we do hereby provide again information that this nucleotide sequence was calculated as 707. Once our work will be published online every reader could trace our strain details at gene bank NCBI.
5. The English language in the second version of the manuscript has not improved. I strongly recommend that the authors contact a native speaker to adjust the style.
Answer: English has been thoroughly checked by British Author in this work and commented English is fine.

Reviewer 2 Report
Comments and Suggestions for Authors
It is not appropriate for publication because few revisions have been made to the points raised.
Author Response
Dear Reviewer,
Thank you for your thoughtful feedback. We, the authors, would like to address the points you raised and here are their point by point answer.
During Phase 1 of your comments following are the comments raised and we answered them and again we answered them to avoid any confusion.
- This paper by Mushtaq et al. on the production of amylase and cellulase using potato peel waste is interesting in terms of environmental issues and industrialization, but is highly deficient in its structure as a scientific paper. Significant revision is required.
Answer. Dear Reviewer, this manuscript has been revised significantly and now in very good structure that could be published as a scientific paper.
2. How much does it cost to use PPW - transport, cleaning, removing all dirt, oven drying, crushing, etc.?
Answer:
Thank you for your thoughtful feedback. We, the authors, would like to address the points you raised:
- Pakistan, a major agricultural country, presents a readily available source of potato peel waste (PPW) due to its widespread potato consumption. The abundance of fries shops, food processing facilities, and restaurants utilizing potatoes generates significant quantities of PPW. Currently, this biowaste is often discarded alongside domestic waste, ultimately reaching landfills. By contrast, our research leverages this readily available and essentially zero-cost resource (collection and drying) for potential value creation.
- Presence of local restaurants across Pakistan minimizes transportation challenges associated with PPW acquisition it’s not any issue in this country. In our study, we conveniently collected potato peels from a local fries shop at Quaid E Azam Campus of Punjab University Gate no 4 in a Shopping Bag for further laboratory analysis. This demonstrates the ease with which this readily available biowaste can be sourced
- Following washing with the water for cleaning and removing the dirt we sundried these peels.
- Following thorough washing with fresh water, the peels were sun-dried for several days, taking advantage of the abundant sunlight. This natural drying approach eliminated the need for additional resources beyond sunlight and water. To ensure complete moisture removal, a final oven drying step for only 6 hours was employed. Overall, the drying process aligns with our research focus on cost-effectiveness, minimizing resource consumption.
- For PPW crushing, we utilized the services of local milling shops readily available throughout Pakistan. These establishments, commonly found in cities, villages, and towns, typically process grain wheat into flour for human consumption. The grinding service for our PPW samples incurred minimal costs, typically less than $1 for processing 10 kg of material. This approach aligns well with the emphasis on cost-effectiveness within our research.
3. Are there any control experiments with or without PPW in the bacteria culture and enzyme assay?
Answer: In this statistical model study, there was no control experiment was conducted neither they are required.
4. Lack of explanations for Figures and Tables. Just by looking at a Figure or Table, one must know what it represents.
Answer: Thank you for your thoughtful feedback. We, the authors, would like to address the points you raised,
- Figure and tables are now updated with very good version and now it represents well.
- Title of all tables are updated with good version
- Title of all the Figures are updated with good version
- Explanation of the figures are already added and now improved with good version
5. Overall, it is not uniform, including the formatting, and is very difficult to read. Also, the results and discussion are poorly worded and it is not clear what they are talking about.
Answer: Dear Reviewer, thank you for highlighting this issue to improve our work, we do hereby describe our position in this regard,
- Manuscript is now uniform with good formatting.
- Manuscript has been improved now and easily readable.
In the results and discussion section, we thoroughly revised whole manuscript. From line 129-135, 218-230, 250-255, 336-340 it has now been updated with very good words, and is clear that we want present cost-effective solution to handle the waste management problem of agricultural and food waste i.e., potato peels. We provided cost effective treatment conditions to optimize the production of most required enzymes in food industry and other sectors
6. Comments on the Quality of English Language Minor editing of English language required.
Answer: Updated with good quality of English language by British author
Phase 2 Comments and their Answer:
It is not appropriate for publication because few revisions have been made to the points raised.
Ans. Dear Reviewer, with extreme apologize in our previous response, The whole paper has been thoroughly checked and reviewed.
- All the points raised by the honorable reviewer are now answered thoroughly, point by point.
- No control experiment was used as in RSM there is no provision of control experiment. Explanations about figures and tables have now been improved.
- All corrections have been made as suggested by learned reviewer.
- There are no answer on the point left behind that is raised by the honorable reviewer.

Reviewer 3 Report
Comments and Suggestions for Authors
I have no further questions for this manuscript.
Author Response
Honorable Reviewer, Many thanks for your time for peer review our manuscript.
We pay sincere gratitude towards you.